# Comparisons and Associations between Hip-Joint Position Sense and Glycosylated Hemoglobin in Elderly Subjects with Type 2 Diabetes Mellitus—A Cross-Sectional Study

**DOI:** 10.3390/ijerph192315514

**Published:** 2022-11-23

**Authors:** Faisal Asiri, Ravi Shankar Reddy, Bayapa Reddy Narapureddy, Abdullah Raizah

**Affiliations:** 1Department of Medical Rehabilitation Sciences, College of Applied Medical Sciences, King Khalid University, Abha 61421, Saudi Arabia; 2Department of Public Health, College of Applied Medical Sciences, King Khalid University, Abha 61421, Saudi Arabia; 3Department of Orthopaedics, College of Medicine, King Khalid University, Abha 61421, Saudi Arabia

**Keywords:** hip joint, proprioception, type 2 diabetes mellitus, joint position sense

## Abstract

Hip-joint position sense (JPS) accuracy may be impaired in individuals with type 2 diabetes mellitus (T2DM). An impaired hip JPS can alter postural control and bodily balance. The objectives of this study are to (1) compare the hip JPS between T2DM and asymptomatic and (2) assess the relationship between hip JPS and glycosylated hemoglobin (HbAlc). This comparative cross-sectional study included 117 elderly individuals with T2DM (mean age: 59.82 ± 6.80 y) and 142 who were asymptomatic (mean age: 57.52 ± 6.90 y). The hip JPS was measured using a digital inclinometer. The individuals were repositioned to a target position with their eyes closed, and the magnitudes of matching errors were estimated as reposition errors. The hip JPS was evaluated in the flexion and abduction directions. The magnitude of reposition errors was significantly larger in the T2DM group in the right flexion (*p* < 0.001), the right abduction (*p* < 0.001), the left flexion (*p* < 0.001), and the left abduction (*p* < 0.001) directions compared to the asymptomatic group. HbA1c values showed a significant positive correlation with JPS in the right-hip flexion (r = 0.43, *p* < 0.001), the right-hip abduction (r = 0.36, *p* < 0.001), the left-hip flexion (r = 0.44, *p* < 0.001), and the left-hip abduction (r = 0.49, *p* < 0.001) directions. Hip JPS testing may be considered when assessing and formulating treatment strategies for individuals with type 2 diabetes. Future research should focus on how hip JPS can impact balance and falls in individuals with T2DM.

## 1. Introduction

Proprioception is critical for maintaining awareness of joint position in both conscious and unconscious states to prevent injury [1]. Proprioception is a term that includes joint position sense (JPS), kinesthesia (perception of active and passive motion), and a sense of tension or force [2]. Impaired JPS has been documented in individuals with hip injuries [3,4,5]. The JPS depends on efficient afferent proprioceptive input from the periphery [6]. An impaired mechanoreceptor’s ability to send afferent input can impact proprioceptive acuity [7,8,9].

Type 2 diabetes mellitus (T2DM), one of the most common metabolic disorders, is caused by a combination of two primary factors: defective insulin secretion by pancreatic β-cells and the inability of insulin-sensitive tissues to respond appropriately to insulin [10]. HbA1c value measures blood-glucose control, which reflects an individual’s average blood-glucose level over two to three months [11]. In healthy individuals, the typical range for HbA1c is 4 to 5.6% (20 to 38 mmol/mol); values between 5.7% and 6.4% (39 to 46 mmol/mol) suggest prediabetes, and HbA1c levels above 6.5% (47 mmol/mol) may indicate diabetes [11]. T2DM is a progressive condition that impairs distal blood flow, leading to musculoskeletal complications such as reduced range of motion, muscle strength, and endurance at the hip joint [12,13]. Periarticular tissues are affected due to the accumulation of glycation end products leading to crosslinking collagen in tissues [14,15]. These changes may impact muscles, tendons, ligaments, and joint cartilage, which can alter the JPS [15]. As foot sensory neuropathy progresses in individuals with T2DM, these changes may lead to alterations in JPS, postural control, and stability [16].

T2DM causes substantial central and peripheral nervous system disturbances [17]. Neuropathic changes can impair muscle, joint, and cutaneous mechanoreceptors [16]. Individuals with neuropathy have an increased risk of musculoskeletal disorders and have 15 times increased risk of falls [18]. As the diabetic nerve is hypoxic, the afferent fibers that transmit kinesthesia and JPS to the higher centers through the spinal cord are significantly affected [17,19]. The hip is one of the most dynamic joints in the body, and hip-joint proprioception is essential for maintaining static and dynamic stability and optimal lower extremity function [20]. Altered hip JPS may greatly affect postural control and lead to gait disturbances and falls in these individuals [3,21]. The hip JPS assessments have gained importance in contemporary clinical practice when evaluating individuals with different hip conditions. It has been established that individuals with T2DM have impaired proprioception [22,23], and there are limited studies examining the hip JPS in this group. Therefore, it is essential to evaluate hip JPS and its relationships with glycated hemoglobin in order to better comprehend and develop rehabilitation strategies. Furthermore, with a greater understanding of how hip JPS is affected in adults with diabetes, a customized exercise program can be devised to improve this client group’s mobility status. 

It is currently unknown if individuals with T2DM may have impaired hip JPS compared to healthy individuals. Furthermore, it is unknown whether there is a relationship between HbA1c values and hip JPS. Therefore, the objectives of this study are (1) to compare the hip JPS among individuals with T2DM and asymptomatic and (2) to assess the relationship between glycated hemoglobin and hip JPS in individuals with T2DM. 

## 2. Materials and Methods

### 2.1. Subjects

One hundred and seventeen elderly individuals with T2DM (mean age: 59.82 ± 6.80 y) and 142 asymptomatic (mean age: 57.52 ± 6.90 y) were enrolled in this cross-sectional study using the convenience sampling method from neighborhood community centers. The study was conducted in the physiotherapy department of King Khalid University in Saudi Arabia, and the data were collected from May 2021 to April 2022. All the individuals were assessed for their eligibility and included if they could walk independently and understand and follow the commands of the examiner. The T2DM subjects were excluded if they had peripheral neuropathy as assessed using 5.07/10 g Semmes Weinstein monofilament examination (SWME), recent foot lesions, visual impairment or retinopathy, unstable hypertension, or a history of neurological disorders. Asymptomatic subjects were included if they were healthy, understood, and followed therapists’ commands. This study followed the Declaration of Helsinki guidelines. Institutional ethical committee approval was obtained from the local university (ECM #2021-6010), and all the subjects provided written informed consent before the commencement of this study. Of the 186 T2DM individuals screened, 69 were excluded, as 28 had peripheral neuropathy, 26 were unwilling to participate, six had foot ulcers, and nine were not able to follow commands. In asymptomatic participants, 62 out of 206 were excluded because 36 had symptoms of osteoarthritis and 26 were unwilling to participate.

### 2.2. Semmes Weinstein Monofilament Examination (SWME)

For detecting light touch pressure sensation assessment, the nylon Semmes–Weinstein monofilaments test (SWMT) kit (North Coast Medical, San Jose, CA, USA) was used at the sole of the foot in three regions (plantar aspects of the great toe, the third metatarsal, and the fifth metatarsals) [24]. During application, fibers were positioned at a 90° angle to the skin, and the duration of the pressure was between one and two seconds. The smallest filament that the patient managed to feel was recorded. The patients who had the inability to perceive the application of 5.07 monofilament were determined to have large fiber neuropathy and were excluded from the study [24,25].

### 2.3. Laboratory Testing for HbA1c

HbA1c was measured using a high-performance liquid chromatography assay [26]. A healthcare expert used a needle to draw blood from one of the veins of the T2DM individuals. HbA1c analysis in blood offers information on an individual’s average blood-glucose levels during the last two to three months, corresponding to the red blood cell half-life [27]. HbA1c levels are expressed in percentage as per the Diabetes Control and Complications Trial units [28]. HbA1c levels ≥ 6.5% are considered diabetic [28].

### 2.4. Assessment of Hip-Joint Position Sense

Hip JPS was assessed by measuring the reposition accuracy of actively positioning the hip to the target position using a dual digital inclinometer (J-tech, Inc. Salt Lake City, UT, USA). The principles of assessing hip JPS were adopted from Reddy et al. [29]. The digital inclinometer is a reliable (ICC  =  0.96, SEM  =  0.04) and valid (ICC  =  0.98, SEM  =  0.08) tool in assessing joint position sense [30]. The target position is set as 25 degrees of hip flexion and 25 degrees of hip abduction. All the evaluations are performed in a well-ventilated and quiet environment. All the participants wore spandex shorts, and the testing was performed standing with their eyes closed. The inclinometer’s position was changed between flexion and abduction direction. The inclinometer was placed and secured at the middle and lateral aspect of the participant’s thigh to measure hip JPS in flexion and in the anterior and middle aspect of the thigh to measure JPS in the abduction direction (Figure 1). 

Participants were instructed to stand on a 15 cm step height on the non-testing leg, allowing the testing leg to flex or abduct freely at the hip joint. The participants were asked to hold a wooden frame in front of them for added support. The examiner guided the participant’s testing leg from the neutral position (0 degrees of the hip) to the 25 degrees of hip flexion or 25 degrees of hip abduction (target position). The examiner held the participant’s leg in the target position for five seconds and asked them to memorize this position. After five seconds, the participants were asked to move the leg back to the neutral or starting position (0 degrees of the hip). Following this, the examiner asked the individuals to reposition their hips to the target position as accurately as possible at a slow and steady pace. When the participants reached the target position, they indicated by saying “YES.” The reproduced joint angles or the deviation from the target were estimated as the absolute error in degrees. The hip JPS testing was performed on both sides, and the order of testing sides (left and right) and different directions were randomized using a simple chit method. Three trials of JPS testing were performed in each movement direction, and the average of the three values was used to analyze the data. All JPS examinations were administered by a single examiner unaware of the subject’s group affiliation. The examiner is skilled in assessing musculoskeletal conditions and has a decade of post-doctoral experience. An assistant assisted the examiner while they documented the results. During the JPS testing, the examiner gave all participants standard instructions and provided no additional feedback or encouragement. 

### 2.5. Sample-Size Estimation

G*power 3.1 (Universities of Dusseldorf, Germany) was utilized to estimate the sample size for two independent groups and continuous variables [31]. We determined the sample size based on the pilot study data for the primary outcome measure, i.e., the target reposition error (Group A mean = 4.8, Group B mean = 3.6, SD = 2.2), using a power of 0.80 and an alpha value of 0.05. The estimated sample size for each group was 106. 

### 2.6. Statistical Analysis

The Statistical Package for the Social Sciences (SPSS) for Windows, Version 24.0 (IBM Corporation, Armonk, NY, USA) was used to analyze the study data. The study data were checked to see if they followed normative distributions using the Shapiro–Wilk tests, and all of the study data followed a normal distribution. One-way ANOVA compared hip JPS between T2DM and asymptomatic groups. Cohen’s d (effect size) is calculated using the formula $\frac{\left(M2-M1\right)}{\;SD\;pooled}$ [32]. M2–M1 is the mean difference between the two groups. The MDC is calculated as the standard error of measurement (SEM) X 1.96 X √2. SEM is calculated using the formula: $SD\left(\sqrt{1}-R\right)$. Pearson’s correlation coefficient (r) assessed correlations between HbA1c and hip JPS in flexion and abduction. The magnitude of (r) was qualitatively assessed as follows: small, 0.1 to <0.3; moderate, 0.3 to <0.6; good, 0.6 to 1. A *p*-value of less than or equal to 0.05 was considered significant.

## 3. Results

This was a cross-sectional study without intervention. Figure 2 shows the CONSORT flow chart of the study.

The baseline and the demographic characteristics of the study population are shown in Table 1. 

Hip JPS was impaired in the T2DM group compared to the asymptomatic group, as summarized in Table 2. 

The magnitude of errors was significantly larger in the T2DM group in the right flexion (*p* < 0.001), the right abduction (*p* < 0.001), the left flexion (*p* < 0.001), and the left abduction (*p* < 0.001) directions compared to the asymptomatic group. The errors in the T2DM group were largest in left-hip abduction (5.48°) and were least in right-hip flexion (5.10°) directions. The position-sense errors in the asymptomatic group were highest in right-hip flexion (3.35°) and were lowest in left-hip abduction (2.35°) directions (Table 2). The SEM ranged from 0.64 to 0.68 degrees, and the MDC ranged from 1.76 to 1.87 (Table 2).

The relationship between HbA1c and hip JPS is summarized as a scatterplot in Figure 3. 

HbA1c values showed a significant positive correlation with JPS in the right-hip flexion (r = 0.43, *p* < 0.001), right-hip abduction (r = 0.36, *p* < 0.001), left-hip flexion (r = 0.44, *p* < 0.001), and left-hip abduction (r = 0.49, *p* < 0.001) directions. As the HbA1c % increased, the hip JPS impaired, indicating that the greater the glycated hemoglobin levels, the worse the hip JPS.

## 4. Discussion

This study aimed to compare hip JPS between the T2DM and asymptomatic groups and to assess the relationship between HbA1c values and hip JPS in T2DM individuals. This study showed that hip JPS errors were more prevalent in the T2DM group than in the asymptomatic group. Furthermore, there was a positive relationship between HbA1c values and hip JPS in the T2DM group.

The decreased JPS in the T2DM group patients can be attributed to the effects of diabetes on the nerve, such as hypoxia, small-fiber degeneration, and altered nociceptive feedback, which may alter the afferent proprioceptive input and impair the joint position sense [33,34]. A prior study by Ettinger et al. demonstrated that individuals with T2DM had impaired knee JPS compared to healthy individuals [35]. The T2DM individuals were 46% less accurate in matching the target angle [35]. There is a paucity of studies examining the proprioceptive responsiveness of the hip joint in individuals with type 2 diabetes, but there are several examining other peripheral joints [35,36,37,38]. The hip JPS declines observed in this study were consistent with previous reports presented by Maras et al. [38] in the ankle joint and by Haghighi et al. [37] in the knee joint. The mean absolute ankle and knee reposition errors were higher in T2DM individuals than in asymptomatic individuals [38]. Similar to our study results, Reddy et al. [22] aimed to compare cervical JPS between individuals with T2DM and healthy individuals and to assess the correlation between HbA1c values and cervical JPS in individuals with T2DM. The participants with T2DM showed significantly larger cervical joint reposition errors in all directions compared with the healthy group (flexion: d = 1.23, *p* < 0.001; extension: d = 1.85, *p* < 0.001; left rotation: d = 1.70, *p* < 0.001; and right rotation: d = 2.60, *p* < 0.001). HbA1c showed a significant positive moderate correlation with cervical JPS in flexion (r = 0.41, *p* = 0.001), extension (r = 0.48, *p* < 0.001), left rotation (r = 0.38, *p* < 0.001), and right rotation (r = 0.37, *p* < 0.001), in participants with T2DM [22]. Proprioceptive information is essential for maintaining balance and preventing falls in the elderly, and reduced JPS in T2DM patients will make these individuals susceptible to falls or injury [39]. Deursen et al. conducted a study on foot and ankle sensory neuropathy. They stated that diabetes impairs the function of peripheral sensory receptors, particularly muscle spindles, which may cause disturbances in balance and gait stability [16]. Patients with diabetes demonstrated decreased reflex responses to postural perturbation, followed by reduced nerve conduction velocity, resulting in balance disturbance and an increased risk of falling [16,40]. The T2DM individuals in this study were overweight, with a BMI of 27.24 compared to the asymptomatic individuals (BMI = 24.58). Previous studies have shown that obese individuals were found to have greater muscle stiffness compared to their lean counterparts [41]. Fatty infiltration of skeletal muscles in obese people may increase muscle stiffness and reduce flexibility compared to non-obese individuals due to the limitation of range of motion and stable posture [42,43]. These changes might have influenced the hip JPS and postural stability in the T2DM individuals. Future studies are warranted to assess the influence of BMI and its magnitude on hip JPS and postural stability. 

Decreased hip-joint proprioceptive acuity can significantly impair postural control and increase the risk of falls [6,44,45,46]. In addition, the effect of age on central mechanics and changes in the peripheral somatosensory system may contribute to JPS impairments in older individuals with T2DM [29,47]. According to studies examining the benefits of proprioceptive exercise regimens on patients with diabetes, balance and postural stability can be improved, most likely by an increase in peripheral inference, resulting in decreased falls caused by sensory deficiencies [48,49,50].

This study is the first to investigate the association between HbA1c and hip JPS in T2DM individuals. The outcomes of this study may provide critical insight into the role of hip JPS testing as a standard diagnostic tool in diabetic patients. Assessing hip proprioception is critical for diabetic patients since the early detection of impaired position sense can prevent falls and other complications in the elderly [39]. This study’s results are in accordance with Khan et al. [51], who showed a positive relationship between HbA1c levels and knee-joint reposition errors (r = 0.570, *p* < 0.001) [51]. Tiwari et al. demonstrated that increased blood-sugar levels resulted in impaired proprioceptive acuity in the knee joint. [52]. Contrary to this study’s results, Maras et al. [38] found no relationship between glycemic control and ankle JPS. Our results cannot be compared to Maras et al.’s study as there are differences in study-method assessments.

Although these findings may not be entirely generalizable to all peripheral joints in the T2DM group, it is still possible to make a number of “best practice” recommendations for how clinically based tests of proprioceptive function should be conducted. Subjects in the study were young (mean age 59.82 years) and physically active; therefore, it has limited generalizability to older individuals (>70 years) and individuals with pathological conditions.

Reduced levels of HbA1c may improve hip JPS. Therefore, future research should investigate the possibility of lowering HbA1c levels and their impact on JPS to gain more relevant data for managing individuals with T2DM.

### Limitations of the Study

We recognize that hip proprioception, as measured by a target repositioning task, has a memory component (memorizing the movement of the leg) that may have influenced this study’s findings. This study estimated hip JPS as absolute errors; constant and variable errors could have provided additional information about the direction (over and under-shooting of the targets) and magnitude of errors.

## 5. Conclusions

T2DM individuals showed impaired hip JPS (increased errors) compared to asymptomatic individuals. Glycated hemoglobin levels showed a positive relationship with hip JPS. This impaired proprioceptive sensibility may be related to the widespread degeneration of afferent nerves in the T2DM population. Following current clinical practice guidelines, hip JPS tests should be considered when assessing and managing individuals with type 2 diabetes. This novel field requires additional research to create evaluation methodologies and appropriate treatment plans for individuals with type 2 diabetes.

## Figures and Tables

**Figure 1 ijerph-19-15514-f001:**
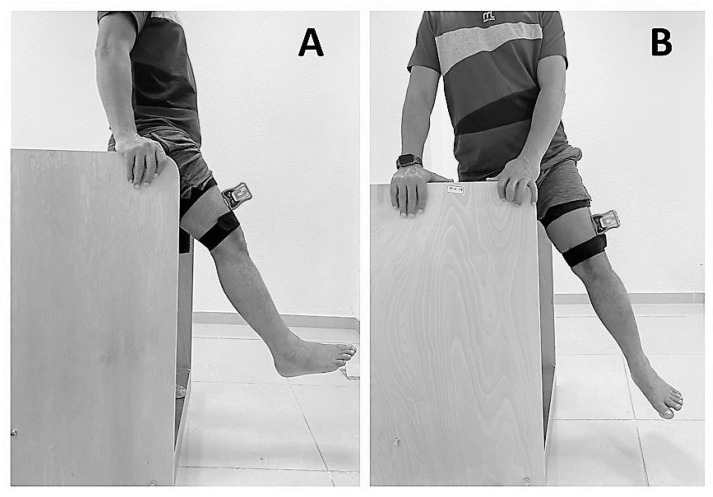
Assessment of hip-joint position sense in standing position (**A**) hip flexion; (**B**) hip abduction.

**Figure 2 ijerph-19-15514-f002:**
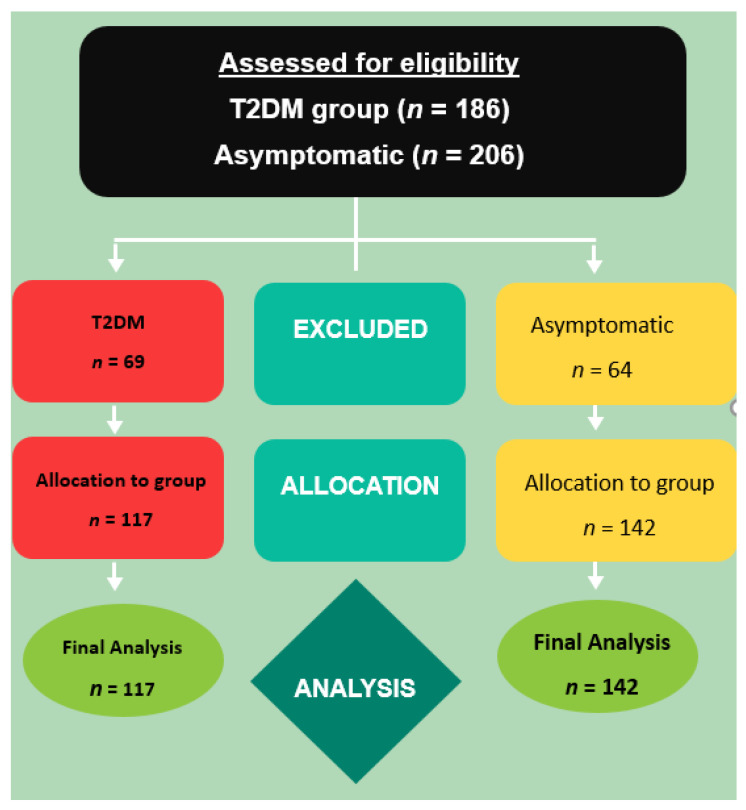
CONSORT flow diagram of the cross-sectional study.

**Figure 3 ijerph-19-15514-f003:**
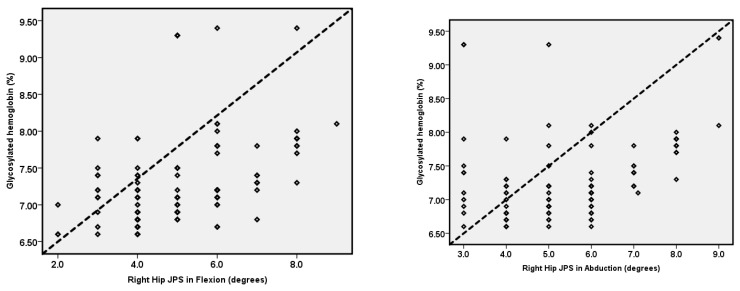
Relationship between HbA1c values and hip JPS in right- and left-hip flexion and abduction.

**Table 1 ijerph-19-15514-t001:** Demographic characteristics of the study population.

Variables	T2DM Group (*n* = 117)	Asymptomatic Group (142)	*p*-Value
Age (years)	59.82 ± 6.80	57.52 ± 6.90	0.065
Gender (M:F)	73:44	86:56	
Height (cm)	167.81 ± 11.69	162.03 ± 7.48	<0.001
Weight (kg)	75.95 ± 9.83	64.40 ± 6.14	<0.001
BMI (kg/m^2^)	27.24 ± 4.63	24.58 ± 2.31	<0.001
Duration of diabetes (years)	6.73 ± 2.33	-	-
Monofilament test (%) Normal Reduced response Absent	73.2 26.8 0	-	-
Glycosylated hemoglobin (HbA1c)	7.29 ± 0.58	-	-

BMI = body mass index.

**Table 2 ijerph-19-15514-t002:** Comparison of hip JPS between T2DM and asymptomatic group.

Variables	T2DM Group (*n* = 117) (Mean ± SD)	Asymptomatic Group (*n* = 142) (Mean ± SD)	95% CI of the Difference		Cohen’s d	SEM	MDC	*p*-Value
			Lower	Upper				
Right-hip JPS in 25° flexion (°)	5.10 ± 1.52	3.35 ± 1.27	1.41	2.09	1.24	0.67	1.85	<0.001
Right-hip JPS in 25° abduction (°)	5.48 ± 1.54	3.01 ± 1.44	2.10	2.83	1.65	0.68	1.87	<0.001
Left-hip JPS in 25° flexion (°)	5.21 ± 1.51	3.08 ± 1.23	1.79	2.46	1.54	0.66	1.82	<0.001
Left-hip JPS in 25° abduction (°)	5.71 ± 1.47	2.35 ± 1.06	3.05	3.67	2.62	0.64	1.76	<0.001

JPS = joint position sense, T2DM = type 2 diabetes mellitus, CI = confidence Interval, SEM = standard error of measurement, and MDC = minimal detectable change. *p* values are based on post-hoc Bonferroni correction.

## Data Availability

All data generated or analyzed during this study are included in Appendix A.

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
