# Peer review of "Comparisons and Associations between Hip-Joint Position Sense and Glycosylated Hemoglobin in Elderly Subjects with Type 2 Diabetes Mellitus—A Cross-Sectional Study"

_ijerph, 2022, doi:10.3390/ijerph192315514_

Round 1

Reviewer 1 Report

This paper discuses hip joint position in subjects with type 2 diabetes mellitus. They included 117 patients and 142 healthy subjects. The materials and methods section is confusing, especially "assessment of hip joint position sense" part. The figure of the experimental work is missing and is needed to clarify the procedure. The results section needs to be expanded, and more charts are needed to show the significance of the work. The discussion section needs more explanation to the results!

The manuscript needs a lot of extensive revision before it can be published. (Detailed in the PDF attached)

Author Response

Response to Reviewer 1

Thank you for your effort and time in reviewing our manuscript. The reviewing process has significantly improved the quality of this manuscript. Therefore, I am submitting this "Response to reviewers" document summarizing the changes we made in response to the critiques. In addition, I have highlighted the changes manuscript.

Note: We have responded to each comment in the pdf document provided. The changes are incorporated in the new manuscript submitted.

Pdf document is attached

Reviewer 2 Report

Dear Authors,

    First of all, I think the manuscript entitled: “Comparisons and associations between Hip joint position sense and glycosylated hemoglobin in elderly subjects with Type 2 Diabetes Mellitus – a cross-sectional study” submitted for publication in the International Journal of Environmental Research and Public Health Journal (MDPI) has both clinical and scientific interest.

More specifically:

Ø  The aim of this study was: 1) to compare the hip JPS between T2DM and asymptomatic and 2) to assess the relationship between hip JPS and glycosylated hemoglobin (HbAlc).

Ø  Generally, the paper is well written. The text is clear and easy to read (beneficial for this purpose are the targeted figures and tables* found within the article).

Ø  The number of sample included in the study is satisfactory based on the results of the sample size estimation.

Ø  The conclusions of the manuscript are in accordance with the evidence as well as the arguments presented by the authors.

Ø  The authors address the central question quite well.

Based on the above:

Overall Recommendation: Accept after minor revision.

Comments and Suggestions for Authors:

*Table 1 (Lines 144-145): The authors should delete the spaces between Average & SD values that exist in certain measured parameters that are presented in the specific table.

Author Response

Response to Reviewer comment 2

Thank you for your effort and time in reviewing our manuscript. The reviewing process has significantly improved the quality of this manuscript. Therefore, I am submitting this "Response to reviewers" document summarizing the changes we made in response to the critiques. In addition, I have highlighted the changes manuscript.

Reviewer 2

Sl.no

Queries

Response to queries

1

first of all, I think the manuscript entitled: “Comparisons and associations between Hip joint position sense and glycosylated hemoglobin in elderly subjects with Type 2 Diabetes Mellitus – a cross-sectional study” submitted for publication in the International Journal of Environmental Research and Public Health Journal (MDPI) has both clinical and scientific interest.

More specifically:

Ø  The aim of this study was: 1) to compare the hip JPS between T2DM and asymptomatic and 2) to assess the relationship between hip JPS and glycosylated hemoglobin (HbAlc).

Ø  Generally, the paper is well written. The text is clear and easy to read (beneficial for this purpose are the targeted figures and tables* found within the article).

Ø  The number of sample included in the study is satisfactory based on the results of the sample size estimation.

Ø  The conclusions of the manuscript are in accordance with the evidence as well as the arguments presented by the authors.

Ø  The authors address the central question quite well.

Thank You for your comments

Overall Recommendation: Accept after minor revision.

Comments and Suggestions for Authors:

*Table 1 (Lines 144-145): The authors should delete the spaces between Average & SD values that exist in certain measured parameters that are presented in the specific table.

We deleted the spaces between mean and SD values in table 1 and 2.

Reviewer 3 Report

This study is mostly about the hip joint position sense in the elderly population with Type 2 Diabetes Mellitus. Even though the article is well written and well designed, I have some comments.  

·         The introduction is good and leaves the reader to the main point of the study but it can be better to give more information or comparison about joint position sense and Type 2 Diabetes Mellitus based on literature.

·         The authors should explain more about the monofilament assessment. Where are the reference points of this test? Why did the authors choose the 10g monofilament or why not the others (Please read; Martinez-Hervás, S., Mendez, M. M., Folgado, J., Tormos, C., Ascaso, P., Peiró, M., ... & Ascaso, J. F. (2017). Altered Semmes–Weinstein monofilament test results are associated with oxidative stress markers in type 2 diabetic subjects. Journal of translational medicine, 15(1), 1-8.)

·         Did the authors give attention to hip joint problems (osteoarthritis etc.) for selecting participants? Moreover, Did the participants use any medication (AINS, analgesics, etc?) during the study? Please expand the inclusion and exclusion criteria of the study

·         The drops of both groups were given in Figure 2. What are the reasons of exclusion, please mention them under the heading of Subjects.

·         The authors must provide more information about the HbA1c sampling. Which method is used for detection?

·         The results are clear and well presented

·         The discussion can be expanded. The authors concluded the elderly population with Type 2 Diabetes Mellitus demonstrated low position sense on both legs and discussed over the peripheral somatosensory system. In this part, the authors may explain how Diabetes Mellitus affects the muscle system because the BMI was found different from than control group. Did the BMI difference between groups influenced your results? Because there can be some adaptation in muscle and joint mechanics in the human body, please read these articles;

o   “Usgu, S., RamazanoÄŸlu, E., & Yakut, Y. (2021). The Relation of Body Mass Index to Muscular Viscoelastic Properties in Normal and Overweight Individuals. Medicina, 57(10), 1022”

o   “Yümin, E. T., ÅžimÅŸek, T. T., Sertel, M., & Ankaralı, H. (2016). The effect of age and body mass index on plantar cutaneous sensation in healthy women. Journal of physical therapy science, 28(9), 2587-2595”

Author Response

Response to Reviewer comments

Thank you for your effort and time in reviewing our manuscript. The reviewing process has significantly improved the quality of this manuscript. Therefore, I am submitting this "Response to reviewers" document summarizing the changes we made in response to the critiques. In addition, I have highlighted the changes manuscript.

Reviewer 3

Sl.no

Queries

Response to queries

1.      

This study is mostly about the hip joint position sense in the elderly population with Type 2 Diabetes Mellitus. Even though the article is well written and well designed, I have some comments.  

·      Thank You for your comments

2.      

The authors should explain more about the monofilament assessment. Where are the reference points of this test? Why did the authors choose the 10g monofilament or why not the others (Please read; Martinez-Hervás, S., Mendez, M. M., Folgado, J., Tormos, C., Ascaso, P., Peiró, M., ... & Ascaso, J. F. (2017). Altered Semmes–Weinstein monofilament test results are associated with oxidative stress markers in type 2 diabetic subjects. Journal of translational medicine, 15(1), 1-8.)

·      Thank you for your advice

·      We have added the information on the monofilament assessment and the reference points tested.

·      There is great variation in the current literature regarding the diagnostic value of Semmes-Weinstein monofilaments examination (SWME) as a result of different methodologies.

·      We followed the guidelines of systematic review, which evaluated current evidence in the literature on the efficacy of Semmes Weinstein monofilament examination (SWME) in diagnosing diabetic peripheral neuropathy.

·      Of the 764 studies identified, 30 articles were selected, involving 8365 patients. There was great variation in both the reference test and the methodology of SWME. However, current literature suggests that nerve conduction study (NCS) is the gold standard for diagnosing DPN. Four studies were identified that directly compared SWME with NCS and encompassed 1065 patients with, and 52 patients without diabetes mellitus. SWME had a sensitivity ranging from 57% (95% confidence interval [CI], 44% to 68%) to 93% (95% CI, 77% to 99%), specificity ranging from 75% (95% CI, 64% to 84%) to 100% (95% CI, 63% to 100%), positive predictive value (PPV) ranging from 84% (95% CI, 74% to 90%) to 100% (95% CI, 87% to 100%), and negative predictive value (NPV) ranging from 36% (95% CI, 29% to 43%) to 94% (95% CI, 91% to 96%).

·      To maximize the diagnostic value of SWME, a three-site test involving the plantar aspects of the great toe, the third metatarsal, and the fifth metatarsals were used.

3.      

The drops of both groups were given in Figure 2. What are the reasons of exclusion, please mention them under the heading of Subjects.

·      The dropouts are now mentioned under the heading of Subjects.

4.      

The authors must provide more information about the HbA1c sampling. Which method is used for detection?

·      The HbA1c sampling method is mentioned in detail.

5.      

 The results are clear and well presented

·      Thank You

6.      

The discussion can be expanded. The authors concluded the elderly population with Type 2 Diabetes Mellitus demonstrated low position sense on both legs and discussed over the peripheral somatosensory system. In this part, the authors may explain how Diabetes Mellitus affects the muscle system because the BMI was found different from than control group. Did the BMI difference between groups influenced your results? Because there can be some adaptation in muscle and joint mechanics in the human body, please read these articles;

o   “Usgu, S., RamazanoÄŸlu, E., & Yakut, Y. (2021). The Relation of Body Mass Index to Muscular Viscoelastic Properties in Normal and Overweight Individuals. Medicina, 57(10), 1022”

o   “Yümin, E. T., ÅžimÅŸek, T. T., Sertel, M., & Ankaralı, H. (2016). The effect of age and body mass index on plantar cutaneous sensation in healthy women. Journal of physical therapy science, 28(9), 2587-2595”

·      Thank you for your comments. We have discussed the differences in BMI between T2DM and asymptomatic and the possible effect of BMI in T2DM individuals and its influence on hip JPS and postural stability.

Round 2

Reviewer 1 Report

The authors have incorporated all my comments and suggestion appropriately. Thanks